

# Development of a prediction model to estimate the 5-year risk of cardiovascular events and all-cause mortality in haemodialysis patients: a retrospective study

Aihong Zhang[1,2,3], Lemuge Qi[1,4], Yanping Zhang[1], Zhuo Ren[1], Chen Zhao[1], Qian Wang[1], Kaiming Ren[1], Jiuxu Bai[1] and Ning Cao[1]

[1] Department of Blood Purification, General Hospital of Northern Theater Command, Shenyang, Liaoning, China
[2] Department of Nephrology, Xi'an People's Hospital (Xi'an Fourth Hospital), Xi'an, China
[3] Postgraduate College, Dalian Medical University, Dalian, Liaoning, China
[4] Postgraduate College, China Medical University, Shenyang, Liaoning, China

## ABSTRACT

**Background**. Cardiovascular disease (CVD) is a major cause of mortality in patients on haemodialysis. The development of a prediction model for CVD risk is necessary to help make clinical decisions for haemodialysis patients. This retrospective study aimed to develop a prediction model for the 5-year risk of CV events and all-cause mortality in haemodialysis patients in China.

**Methods**. We retrospectively enrolled 398 haemodialysis patients who underwent dialysis at the dialysis facility of the General Hospital of Northern Theater Command in June 2016 and were followed up for 5 years. The composite outcome was defined as CV events and/or all-cause death. Multivariable logistic regression with backwards stepwise selection was used to develop our new prediction model.

**Results**. Seven predictors were included in the final model: age, male sex, diabetes, history of CV events, no arteriovenous fistula at dialysis initiation, a monocyte/lymphocyte ratio greater than 0.43 and a serum uric acid level less than 436 mmol/L. Discrimination and calibration were satisfactory, with a C-statistic above 0.80. The predictors lay nearly on the 45-degree line for agreement with the outcome in the calibration plot. A simple clinical score was constructed to provide the probability of 5-year CV events or all-cause mortality. Bootstrapping validation showed that the new model also has similar discrimination and calibration. Compared with the Framingham risk score (FRS) and a similar model, our model showed better performance.

**Conclusion**. This prognostic model can be used to predict the long-term risk of CV events and all-cause mortality in haemodialysis patients. An MLR greater than 0.43 is an important prognostic factor.

Corresponding authors
Ning Cao, bzxyjhk@126.com
Jiuxu Bai, 107034054@qq.com

## INTRODUCTION

The all-cause mortality risk in the haemodialysis (HD) population is much higher than that in the general population. Cardiovascular disease (CVD) is a major cause of mortality in patients on maintenance haemodialysis (MHD). According to 2020 data from the United States Renal Data System (USRDS), CVD accounts for 55.2% of deaths among haemodialysis patients in the United States (*Johansen et al., 2021*). Moreover, CVD accounts for more than 50% of the deaths among haemodialysis patients in China (*Song et al., 2017*). Therefore, the development of a prediction model for CVD risk is necessary to help make clinical decisions for haemodialysis patients.

The Framingham risk score (FRS) is a widely used tool to predict the 10-year cardiovascular disease risk in the general population and includes traditional cardiovascular risk factors (*e.g.*, age, sex, smoking habit, blood pressure, total cholesterol, low-density lipid cholesterol, high-density lipid cholesterol, and diabetes mellitus (DM)) (*Grundy et al., 1999*). However, the FRS has poor overall accuracy in predicting cardiovascular (CV) events in individuals with chronic kidney disease (CKD) (*Huang et al., 2013*) and cannot fully predict the risk of mortality in the haemodialysis population (*Floege et al., 2015*). The possible reason is that "reverse epidemiology" has been observed in CKD patients (*Kalantar-Zadeh et al., 2003*); for example, serum cholesterol, as a traditional cardiovascular risk factor, is not associated with a reduced survival rate in end-stage kidney disease (ESRD) patients (*Anker et al., 2016*). The use of dialysis-related risk factors for CVD (*e.g.*, dialysis age, vascular access, type of first dialysis, serum calcium, serum phosphorus, serum albumin or total parathyroid hormone) can improve the predictive ability of the risk scores (*Floege et al., 2015*). In recent years, an increasing number of studies have shown that some routine blood indexes, such as red blood cell distribution width (RDW) (*Fukasawa et al., 2017*), monocyte/lymphocyte ratio (MLR) (*Xiang et al., 2018*) and neutrophil/lymphocyte ratio (NLR) (*Yoshitomi et al., 2019*), are independent predictors of cardiovascular events or death in haemodialysis patients and can be used to develop a model.

Although some models that include HD-specific risk factors have been created for the haemodialysis population, the majority were used to predict early death after dialysis initiation (*Inaguma et al., 2019*), had a shorter follow-up time (*Matsubara et al., 2017*), or needed complex or expensive indicators (*Floege et al., 2015*). No simple widely used clinical tool is currently available.

In this retrospective study, we developed a five-year risk score for assessing the risk of CV events or all-cause mortality among MHD patients in China based on traditional cardiovascular risk factors, dialysis-related risk factors, and new inflammatory factors, all of which are simple clinical, dialysis, or routine laboratory parameters. We also performed internal validation of the model and constructed a simple clinical score.

## MATERIALS & METHODS

### Study subjects

This study was designed as an observational cohort study. We retrospectively enrolled all haemodialysis patients who received dialysis at the dialysis facility of the General Hospital of Northern Theater Command in June 2016. The exclusion criteria were as follows: (1) age under 18 years; (2) dialysis vintage less than 6 months; (3) tumour, rheumatic immune disease, liver cirrhosis or chronic obstructive pulmonary disease; (4) surgery or blood transfusion within the previous three months; and (5) infection or cardiovascular events within the previous month. The cohort was followed up to 31 May 2021. We also excluded patients who underwent kidney transplantation or transferred to peritoneal dialysis or to another dialysis centre during the follow-up period, patients who were missing data on the outcomes, and those who experienced cardiovascular events or death before September 1, 2016 (at least 90 days of survival thereafter). Finally, 398 patients were included in the present study (Fig. 1).

The study was performed according to the principles laid down in the Declaration of Helsinki and was approved by the Clinical Research Ethics Committee of the General Hospital of Northern Theater Command, approval number Y2021-145. Because our study was a retrospective data review, the Clinical Research Ethics Committee of the General Hospital of Northern Theater Command did not request written informed consent.

### Data collection

We collected demographic, clinical, dialysis and laboratory data from medical records. The patient characteristics were as follows: age, sex, smoking habit, primary kidney disease, dialysis duration, dialysis frequency, type of vascular access and complications. CV events and death and their occurrence time during the follow-up were also recorded. In our centre, routine medical evaluations are performed. Fasting venous blood samples were collected every three months. We recorded all measurements over the first three months of the study.

### Outcome definition

Cardiovascular (CV) events were defined as acute myocardial infarction, stroke, or peripheral vascular events and sudden cardiac death. Acute myocardial infarction was defined as myocardial ischaemia indicated by symptoms, electrocardiogram or cardiac biomarkers, narrowing/obstruction indicated by coronary angiography, percutaneous coronary intervention or coronary artery bypass surgery (*Kimura et al., 2019*). Stroke was defined as bleeding or infarction on computed tomography (CT) or magnetic resonance imaging (MRI) accompanied by neurological symptoms (*Inaguma et al., 2019*). Peripheral vascular events were defined as the amputation of fingers and limbs secondary to vascular disease, peripheral arterial bypass or angioplasty (*O'Hare & Johansen, 2001*). The composite outcome was defined as CV events and/or all-cause death.

### Statistical analyses

The Kolmogorov–Smirnov test was used to examine the normal distribution of the continuous variables. The normally distributed variables are presented as the means $\pm$ SDs.

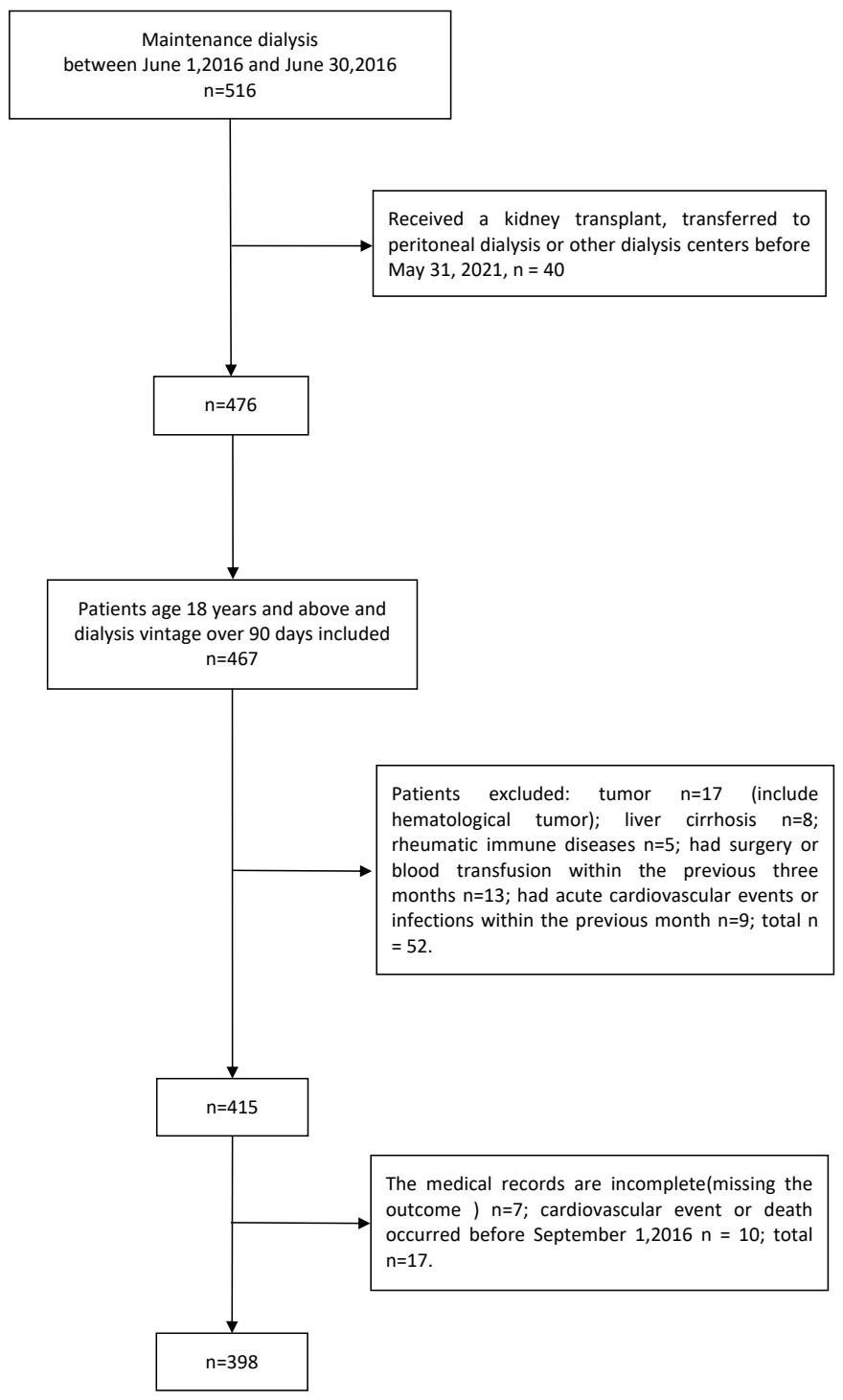

**Figure 1  Flow chart of the participants.**

Independent-samples t test was used to assess the differences in mean values. The nonnormally distributed variables are presented as medians and interquartile ranges (25th percentile, 75th percentile). Comparisons between groups were performed using the Mann–Whitney U test. Categorical variables were described as numbers and percentages, and differences were evaluated using the chi-squared test.

We used item mean imputation to address missing data. Spearman's correlation analysis was used to estimate the presence of collinearity between predictors. Restricted cubic splines were used to estimate nonlinear relationships between the continuous predictors and outcome.

We used multivariable logistic regression with backwards stepwise selection to develop the prediction model. We included predictors considered to have clinical value based on prior reviews or clinical guidelines. We determined the score for each predictor based on the beta coefficient of the resulting model. A scoring system combined with a figure to obtain predicted probabilities was provided for each score in an individual. The C-statistic (area under the receiver operating characteristic (ROC) curve) was used to evaluate the discrimination ability of the model. We constructed a calibration plot and used the Hosmer–Lemeshow test to verify the consistency between the prediction probability and observation probability. Bootstrapping with 1000 resamples was used to assess internal validation. We compared the new model with FRS and a similar model (Matsubara's model) developed in MHD patients from the Japan Dialysis Outcome and Practice Patterns Study (JDOPPS) using the C-statistic, net reclassification improvement (NRI) and integrated discrimination improvement (IDI) (*Matsubara et al., 2017*).

We used IBM SPSS statistics version 26.0 (IBM, Armonk, NY, USA) and R software version 4.1.1 (*R Core Team, 2021*) for all statistical analyses. A $p$ value $< 0.05$ indicated statistical significance. We followed the Transparent Reporting of a multivariable prediction model for Individual Prognosis Or Diagnosis (TRIPOD) statement for reporting (*Moons et al., 2015*).

# RESULTS

## Baseline characteristics

A total of 398 maintenance HD patients were included in our study. The characteristics of all 398 HD patients are shown in Table 1. Among the 398 individuals, 262 (65.8%) were men. The mean age was $53.25 \pm 13.88$ years old. There were 89 (22.4%) individuals with diabetes and 100 (25.1%) with a history of CV events (shown in Table 1).

During the five-year study period, a total of 155 patients reached the composite endpoint. The characteristics and differences between these two groups are shown in Table 1. Men had more outcomes than women. One hundred twenty patients exhibited all-cause mortality, and 35 patients survived CV events.

## Candidate predictors

We selected candidate predictors based on previous studies; these included cardiovascular risk factors (*i.e.,* age, sex, history of DM, causes of renal failure, smoking), dialysis-related risk factors (*i.e.,* vintage, access at dialysis initiation, long-term access, serum calcium,

**Table 1  Baseline characteristics and comparison of patient characteristics and laboratory data.**

| Variable | Overall (n = 398) | Survival without CV events (n = 243) | CV events or all-cause death (n = 155) | P value | Missing (%) |
|---|---|---|---|---|---|
| Age (years) | 53.25 ± 13.88 | 48.50 ± 12.77 | 60.69 ± 12.22 | <0.001 | 0 |
| Male sex (%) | 262 (65.8) | 151 (62.1) | 111 (71.6) | 0.052 | 0 |
| History of DM (%) | 89 (22.4) | 28 (11.5) | 61 (39.4) | <0.001 | 0 |
| History of CV events (%) | 100 (25.1) | 39 (16.0) | 61 (39.4) | <0.001 | 0 |
| Causes of renal failure | | | | <0.001 | 0 |
| CGN | 118 (29.6) | 88 (36.2) | 30 (19.4) | – | 0 |
| DN | 72 (18.1) | 24 (9.9) | 48 (31.0) | – | 0 |
| HT | 48 (12.1) | 34 (14.0) | 14 (9.0) | – | 0 |
| PKD | 21 (5.3) | 9 (3.7) | 12 (7.7) | – | 0 |
| Others or unknown | 139 (34.9) | 88 (36.2) | 51 (32.9) | – | 0 |
| Smoker (yes, %) | 127 (31.9) | 83 (34.2) | 44 (28.4) | 0.229 | 0 |
| Access at dialysis initiation | | | | | 0 |
| AVF (yes, %) | 106 (26.6) | 77 (31.7) | 29 (18.7) | 0.004 | 0 |
| Long-term access | | | | | 0 |
| AVF (yes, %) | 372 (93.5) | 229 (94.2) | 143 (92.3) | 0.436 | 0 |
| Session frequency | | | | | 0 |
| Three times a week (yes, %) | 215 (54.0) | 127 (52.3) | 88 (56.8) | 0.379 | 0 |
| Kt/V | 1.27 ± 0.22 | 1.25 ± 0.26 | 1.30 ± 0.24 | 0.078 | 24.4 |
| Vintage (years) | 4.30 (2.13,7.59) | 4.55 (2.37,8.14) | 3.88 (1.88,6.30) | 0.069 | 0 |
| Serum corrected calcium (mmol/L) | 2.46 ± 0.60 | 2.37 ± 0.54 | 2.61 ± 0.65 | <0.001 | 0.5 |
| Serum phosphorus (mmol/L) | 2.15 ± 0.58 | 2.20 ± 0.56 | 2.06 ± 0.60 | 0.028 | 0.3 |
| Serum albumin (g/L) | 38.65 ± 3.02 | 39.21 ± 2.64 | 37.78 ± 3.35 | <0.001 | 0.5 |
| Haemoglobin (g/L) | 108.45 ± 13.83 | 107.58 ± 13.03 | 108.24 ± 15.00 | 0.808 | 0.3 |
| Serum uric acid (mmol/L) | 476.72 ± 94.94 | 491.05 ± 92.55 | 454.26 ± 94.57 | <0.001 | 0.8 |
| iPTH (pg/ml) | 286.75 (136.50,557.38) | 332.00 (180.45,620.00) | 226.00 (107.55,407.50) | <0.001 | 1.8 |
| MLR | 0.337 (0.259,0.438) | 0.333 (0.258,0.400) | 0.370 (0.261,0.500) | 0.001 | 0.3 |
| NLR | 3.49 (2.75,4.50) | 3.38 (2.79,4.32) | 3.73 (2.66,5.14) | 0.048 | 0.3 |
| RDW | 14.50 (13.85,15.35) | 14.30 (13.65,15.15) | 14.75 (14.05,15.70) | <0.001 | 0.3 |

**Notes.**

Continuous variables are expressed as the means ± SD for normally distributed data and medians (25th percentile -75th percentile) for skewed data. Categorical variables are expressed as percentages (%).

AVF, arteriovenous fistula; CGN, chronic glomerulonephritis; CV events, cardiovascular events; DM, diabetes mellitus; DN, diabetic nephropathy; HT, hypertensive nephropathy; iPTH, intact parathyroid hormone; MLR, monocyte/lymphocyte ratio; NLR, neutrophil/lymphocyte ratio; PKD, polycystic kidney disease; RDW, red blood cell distribution width; –, not applicable.

Formula: Serum corrected calcium (mmol/L) = measured calcium (mmol/L) + (40 − measured albumin (g/L)) ×0.2.

serum phosphorus, intact parathyroid hormone, haemoglobin, serum albumin, serum uric acid (SUA), Kt/V and nontraditional risk factors, as inexpensive markers are included in the complete blood count (CBC) test (*i.e.,* RDW, NLR, and MLR). We performed Spearman's correlation analysis on the following clinically related predictors: (1) history of DM and causes of renal failure and (2) MLR, RDW and NLR. We observed collinearity in the factors above and selected one of the predictors in each group—history of DM and MLR—to develop the model (Table S1). Finally, we chose 11 predictors: age, sex, history

**Table 2 Prognostic Model from the Multivariable logistic regression analysis.**

| Variable | beta | OR (95% CI) | *P* value | Points |
|---|---|---|---|---|
| Age, years | 0.054 | 1.056 (1.033,1.079) | <0.001 | – |
| <55 | – | – | – | 0 |
| 55–64 | – | – | – | 1 |
| 65–74 | – | – | – | 2 |
| ≥75 | – | – | – | 4 |
| Male, % | 0.755 | 2.127 (1.242,3.641) | 0.006 | 1 |
| History of DM, % | 1.307 | 3.694 (2.042,6.682) | <0.001 | 2 |
| History of CV events, % | 0.613 | 1.846 (1.048,3.251) | 0.034 | 1 |
| AVF at dialysis initiation, % | | | | |
| No | 0.839 | 2.315 (1.302,4.117) | 0.004 | 2 |
| Yes | | Reference | | 0 |
| MLR | | | | |
| <0.43 | | Reference | | 0 |
| ≥0.43 | 0.839 | 2.314 (1.328,4.033) | 0.003 | 2 |
| SUA, mmol/L | | | | |
| <436 | 0.802 | 2.231 (1.303, 3.821) | 0.003 | 1 |
| ≥436 | | Reference | | 0 |
| Intercept | −5.515 | 0.004 | <0.001 | – |

**Notes.**

AVF, arteriovenous fistula; CV events, cardiovascular events; DM, diabetes mellitus; MLR, monocyte/lymphocyte ratio; SUA, serum uric acid; –, not applicable.

of DM, history of CV events, vascular access at dialysis initiation, serum albumin, serum corrected calcium, serum phosphorus, intact parathyroid hormone, SUA and the MLR.

The nonlinear relationships between these predictors and the outcome were estimated using restricted cubic splines. After the assessment, the continuous predictor of age showed a good linear relationship with the outcome. Nonlinear relationships were found for serum albumin, serum corrected calcium, serum phosphorus, intact parathyroid hormone, SUA and the MLR. We transformed these continuous variables with nonlinear relationships into categorical variables, as shown in Table 2. The conversion of serum corrected calcium, serum phosphorus and intact parathyroid hormone was based on prior literature (*Eknoyan, Levin & Levin, 2003*), and serum albumin, SUA and the MLR were classified according to the recommended cut-off value (MLR = 0.43, serum albumin = 38.5 g/L, SUA = 436 mmol/L) suggested by the ROC curves.

## Model development and risk score derivation

We conducted a univariate logistic analysis of the selected risk factors. Univariate logistic analysis identified that age, a history of DM, a history of CV events, access at dialysis initiation that was not an arteriovenous fistula (AVF), the serum albumin level, the serum corrected calcium level, the serum phosphorus level, the intact parathyroid hormone level, the SUA level and the MLR were significantly associated with the outcome ($p < 0.05$) (Table 1).

We included 11 factors in our initial model. A logistic model predicting CV events or all-cause mortality was created using a backwards stepwise procedure and included only seven selected predictors: age (odds ratio (OR): 1.056; $p < 0.001$), male sex (OR: 2.127; $p = 0.006$), diabetes (OR: 3.694; $p < 0.001$), CV events (OR: 1.846; $p = 0.034$), no AV fistula at dialysis initiation (OR: 2.315; $p = 0.004$), MLR $\geq 0.43$ (OR: 2.314; $p = 0.003$), and SUA $< 436$ mmol/L (OR: 2.231; $p = 0.003$) (Table 2).

Every additional 10 years in age was equivalent to a one-point increase in the beta coefficient (Wilson et al., 1998), and the total score ranged from 0 to 13 points (Table 2). The number of individuals associated with each total score is shown in Fig. S1. A simple score sheet combined with a figure that was used to obtain predicted probabilities for each score for an individual is shown in Fig. 2. According to the quartile of risk, the score was divided into four groups (low risk: 0–2 points, medium risk: 3–4 points, high risk: 5–6 points, and very high risk: 7–13 points); the risk of each group (low risk: $< 12.4\%$, medium risk: 12.4%–29.5%, high risk: 29.5%–55.2%, and very high risk: $\geq 55.2\%$) is shown in Fig. 2. The incidence of observed CV events or all-cause mortality in the risk groups is shown in Fig. S2.

### Discrimination and calibration of the model

The C-statistic for the composite endpoint was 0.824 (95% confidence interval (CI), 0.783–0.866) in the model (shown in Fig. 3). We constructed a calibration plot (shown in Fig. 4) that showed good agreement between the expected and observed outcomes. We also used the Hosmer–Lemeshow test to verify the consistency between the prediction probability and observation probability. The Hosmer–Lemeshow test results were $X^2 = 3.167$, $P = 0.923$, indicating that the model had good accuracy. The discrimination ability of the new risk model for the incidence of all-cause mortality was also good (AUC: 0.836, 95% CI [0.794–0.879]; Fig. S3).

We compared the performance of the prediction model before and after imputation. The C-statistic of our prediction model on our date before imputation was 0.824 (95% CI [0.782–0.866]), and the ROC curve is shown in Fig. S4. The C-statistic is very similar to our model (C-statistic: 0.824, 95% CI [0.783–0.866]).

### Comparison with Matsubara's model and FRS

The ROC curves of our model and Matsubara's model for the composite outcome are shown in Fig. 3. For Matsubara's model, the C-statistic was 0.767 (95% CI [0.718–0.817]). Our model showed better performance than Matsubara's model. Significant differences were found between our model and Matsubara's model in NRI (2.51%; $p < 0.001$) and IDI (8.45%; $p < 0.001$). Our new model showed improvement in discrimination and the average sensitivity and specificity to predict CV events and all-cause mortality.

The C-statistic was 0.732 (95% CI: 0.646, 0.819) for women (Fig. S5A) and 0.714 (95% CI [0.650–0.777] for men (Fig. S5B) in the FRS. Both our model and Matsubara's model have better performance than the FRS.

| Risk factor | Addition to risk score | | | | Risk score |
|---|---|---|---|---|---|
| Age (year) | **<55** 0 | **55-64** +1 | **65-74** +2 | **≥75** +4 | |
| SUA (mmol/L) | **<436** +1 | **≥436** 0 | | | |
| MLR | **<0.43** 0 | **≥0.43** +2 | | | |
| Sex | Female 0 | Male+1 | | | |
| Diabetes | No 0 | Yes+2 | | | |
| History of CV events | No 0 | Yes+1 | | | |
| AVF at dialysis initiation | No +2 | Yes 0 | | | |
| | | | | Total risk score= | |

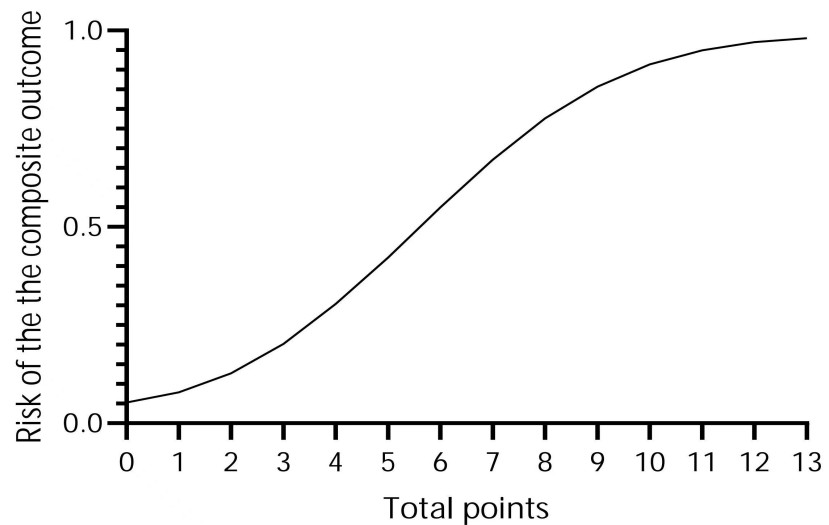

| Categories | low risk | Medium risk | High risk | Very high risk |
|---|---|---|---|---|
| Risk | <12.4% | 12.4%-29.5% | 29.5%-55.2% | ≥55.2% |

**Figure 2** **The score sheet for CV events or all-cause mortality in maintenance haemodialysis patients.**
Points along the *x*-axis of the plot correspond to the approximate probability of CV events or all-cause mortality, which is shown along the *y*-axis. The score is divided into four categories according to the risk quartile. Abbreviations: AVF, arteriovenous fistula; CV events, cardiovascular events; MLR, monocyte/-lymphocyte ratio; SUA, serum uric acid.

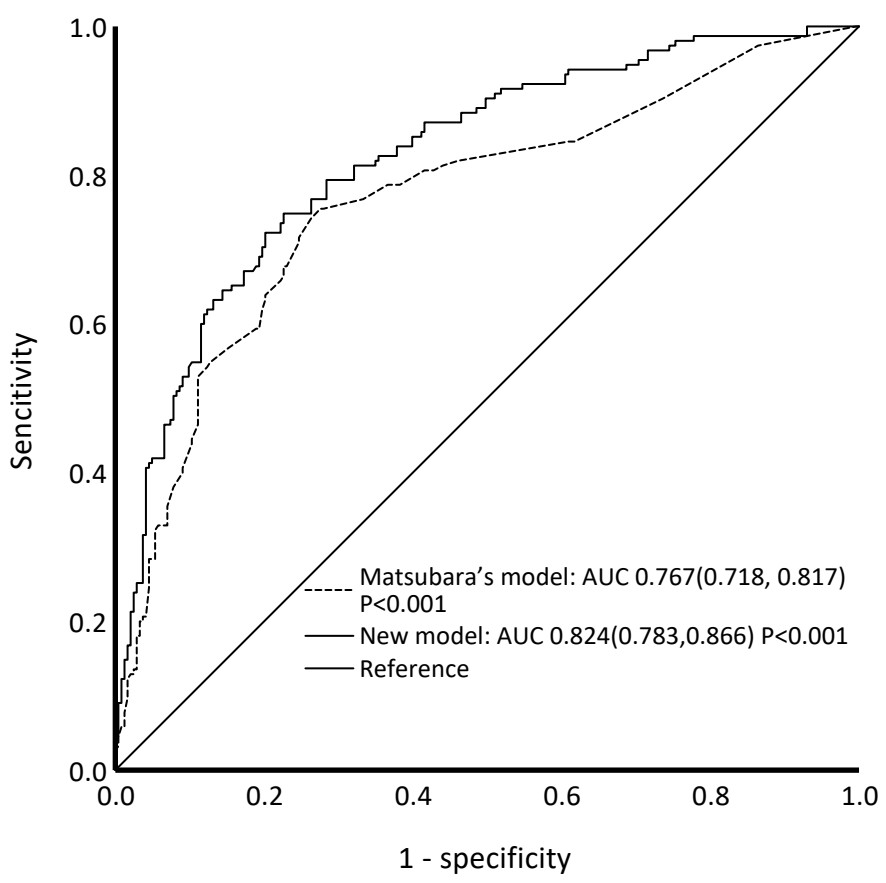

**Figure 3** **ROC curves of our new model and Matsubara's model for the composite outcome.** Our new model showed better performance than Matsubara's model. Abbreviations: ROC, receiver operating characteristic; AUC, area under the curve.

## Internal validation

We used bootstrapping for internal validation and found that the C-statistic (C-statistic: 0.843; 95% CI [0.794–0.876]) and calibration plot in bootstrapping were similar to those of the model of the original cohort (shown in Fig. 5).

## DISCUSSION

In the present study, we developed a prediction score to estimate the risk of CV events and/or all-cause mortality in MHD patients followed up to five years. A total of seven predictors (age, sex, DM, history of CV events, VA type at dialysis initiation, the MLR, and SUA) were included in the model. Our model had good calibration and discrimination, with a C-statistic greater than 0.80. Additionally, the discrimination and calibration were similar to those in the bootstrapping model. The discrimination ability to predict the risk of all-cause death alone is also good (C-statistic: 0.836).

Our model is particularly different from previous models in the following aspects. First, we focused on MHD patients rather than incident HD patients (*Couchoud et al.,*

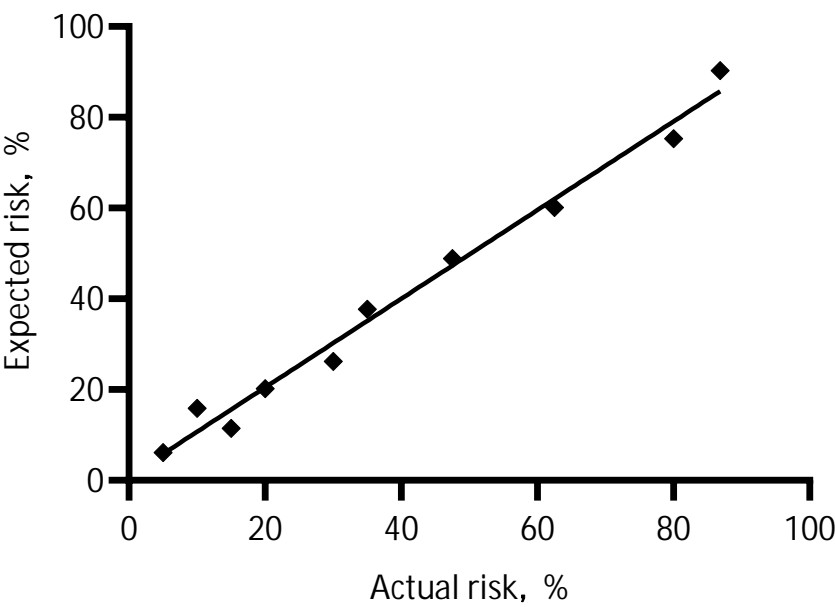

**Figure 4** Calibration of the prediction model—the observed 5-year probability of CV events or mortality by tenths of model-predicted probability.

*2008*; *Inaguma et al., 2019*; *Thamer et al., 2015*; *Wick et al., 2017*). Several predicted early mortality risk scores have been developed, but these scores do not apply to MHD patients whose dialysis duration was more than 6 months because the factors used to develop the scores were collected at the start of haemodialysis. Fluctuations in these predictors after dialysis may be associated with the occurrence of cardiovascular events or all-cause mortality. Second, we included patients aged more than 18 years rather than only older dialysis patients (*Thamer et al., 2015*; *Wick et al., 2017*). The aim of our study was to predict the long-term risk of CV events and all-cause mortality in haemodialysis patients. Although some predictors differ between elderly patients and young people, these factors mainly affect outcomes of the first six months of dialysis initiation. Therefore, previous models for elderly patients are mainly used to predict outcomes within six months at the start of dialysis. Additionally, age was a major predictor in our model, so including both young and old patients will not affect the performance and practicality of our model. Third, our follow-up time of nearly five years is longer than those of most recent prediction models (one year or two years), which were much shorter than that of the FRS (*Anker et al., 2016*; *Matsubara et al., 2017*). Finally, our model included some of the same variables used in other risk scores for haemodialysis patients, most of which are traditional cardiovascular risk factors such as age, sex, DM, and history of CV events (*Ma & Zhao, 2017*; *Song et al., 2017*). However, our prediction model included three variables that were not considered in other prediction models, namely, vascular access type at dialysis initiation, the MLR and the SUA. Additionally, our model is more accurate than a similar model by Matsubara developed to predict cardiovascular events in MHD patients and that included age, diabetes

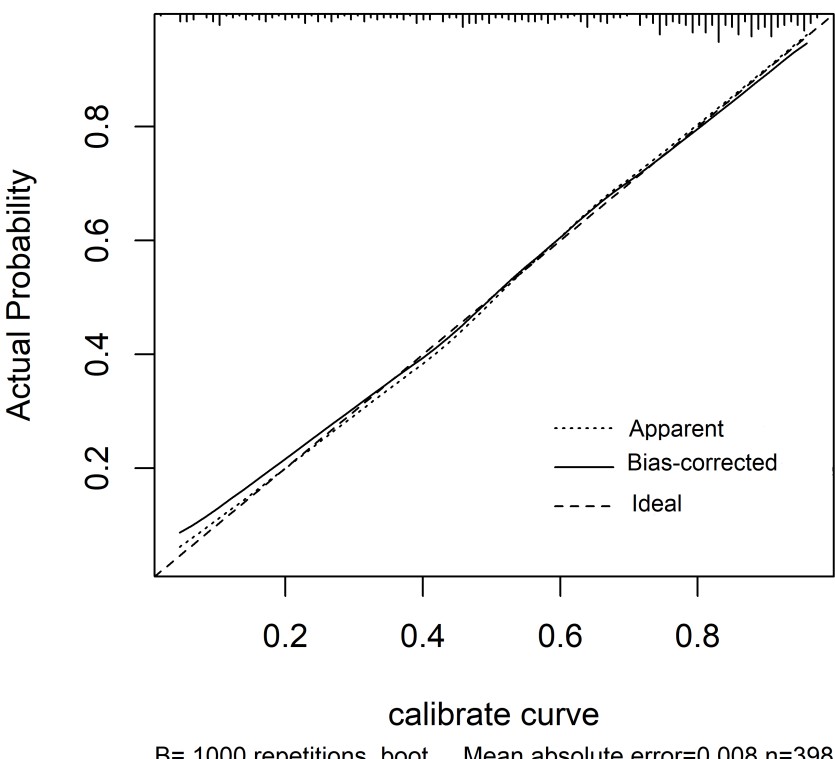

**Figure 5** **Calibration plot for the bootstrapping model.** The line labelled "Apparent" shows the calibration accuracy of the original model. The line labelled "Bias-corrected" shows the relationship between actual and predicted probabilities of the bootstrap model. The line labelled "Ideal" is the reference line. The plot indicates good agreement between the Apparent calibration and the Bias-corrected calibration.

status, history of CV events, dialysis time per session, and serum phosphorus and albumin levels.

Our model is the first to include vascular access (AV) at dialysis initiation as a predictor. A well-functioning VA is the basis of efficient haemodialysis treatment. There are three main types of VA: native AVF, arteriovenous graft, and central venous catheter (CVC). Recent studies have shown that an autologous AVF is still the best choice for VA (*Almasri et al., 2016*) because CVC use places patients at high risk of morbidity and mortality (*Santoro et al., 2014*). AVF use is strongly recommended by guidelines from different countries. The results of our study show that it is consistent with the previous conclusion. Patients who started dialysis with AVF had fewer cardiovascular events and lower rates of all-cause death. Multivariable logistic regression results showed that no AVF at dialysis initiation was an independent risk factor for the composite outcome. Additionally, we found that there was no difference in types of long-term vascular access. This may be because only a very small number of patients use the CVC as long-term vascular access due to the creation of an AVF that fails or cannot be created for different reasons, leading to statistical offset.

The MLR is collected from routine laboratory tests necessary for haemodialysis patients without a corresponding cost increase. Previous studies have suggested that a higher

MLR is an independent risk factor for all-cause mortality and cardiovascular events in HD patients (*Muto et al., 2021*; *Xiang et al., 2018*). Generally, the monocyte count reflects systemic oxidative stress, inflammation, and tissue damage, and the lymphocyte count reflects mostly immune function. In dialysis patients, persistent low-grade inflammation is a feature associated with cardiovascular complications and adverse outcomes (*Snaedal et al., 2016*). Additionally, atherosclerosis is the leading cause of cardiovascular disease, and studies have found that monocyte and macrophage metabolism play a key role in the progression of atherosclerosis (*Groh et al., 2018*). Therefore, the MLR is likely also an inflammation marker and may be a better predictor for cardiovascular morbidity and mortality risk than the absolute monocyte count because it is easily affected by infection, stress, or drugs. The MLR may also perform better than traditional inflammatory markers. CRP (C-reactive protein) is a widely used inflammation marker for cardiovascular events and mortality both in the general population and HD patients (*Ma & Zhao, 2017*). One observational study suggested that an elevated MLR had better predictive value than C-reactive protein (*Gupta et al., 2021*). Additionally, a poor nutritional status in dialysis patients is associated with significantly higher inflammatory marker levels (*Snaedal et al., 2016*). We obtained the same result that patients who achieved the outcomes had a lower serum albumin level and a higher MLR level, although the serum albumin level was not included in the final model. In this study, we proved that the MLR has an excellent linear relation with the NLR. The NLR is also a strong risk factor for cardiovascular mortality (*Li et al., 2017*). *Xiang et al. (2018)* suggested that the MLR is superior to the NLR in HD patients. The addition of the MLR improves the calibration and discrimination of the prediction model.

HD patients have a high prevalence of protein-energy malnutrition and inflammation, along with abnormal iron status. Malnutrition is common in patients and is closely related to morbidity and mortality in haemodialysis (*Marcén et al., 1997*). Univariate logistic analysis identified that the serum albumin level was significantly associated with the outcome in our study. However, the serum albumin level was not included in the final model. Albumin levels, as a marker of nutritional status, are commonly used to evaluate haemodialysis patients and are somewhat criticized because they are affected by inflammation and hydration status (*Hakim & Levin, 1993*). We found that the MLR is likely an inflammation marker and may be a better predictor of cardiovascular morbidity and mortality risk. The most suitable parameters for assessing nutritional status, inflammatory response, and iron status in HD patients have yet to be identified. A recent study found that the use of transferrin levels to assess haemodialysis quality combines into a single test the evaluation of the three most important factors of protein-energy wasting (*Tarantino et al., 2019*). Transferrin may be an ideal candidate factor for the prediction model, although we did not collect relevant data retrospectively.

Hyperuricaemia is a risk factor for CVD and all-cause mortality in the general population. Although hyperuricaemia is common in haemodialysis patients, the relationship between the SUA level and risk of death in haemodialysis patients is controversial. A retrospective study showed a ''J-shaped'' relationship between SUA levels and the risk of cardiovascular death, and high SUA levels increased the risk of cardiovascular death in haemodialysis

patients (*Hsu et al., 2004*). In recent years, an increasing number of studies have shown that high SUA levels may reduce the risk of all-cause death and cardiovascular death in haemodialysis patients (*Bae et al., 2016*; *Dong et al., 2020*; *Zawada et al., 2020*). A meta-analysis showed that each 1 mg/dL increase in the SUA level reduced the risk of all-cause and cardiovascular death by 6% and 9%, respectively (*Wang et al., 2021*). Our study showed that the risk in haemodialysis patients with an SUA level < 436 mmol/L was twice that of patients with an SUA level ≥ 436 mmol/L. The mechanism of the reverse epidemiological phenomenon of the SUA level is not clear at present. Some studies have shown that SUA is related to the nutritional status of haemodialysis patients and may be a nutritional factor (*Bae et al., 2016*). Studies have also suggested that SUA may be a powerful oxygen free radical scavenger and one of the most important antioxidants in human body fluids (*Hsu et al., 2017*).

Methodologically, we included continuous variables with linear relationships in the multivariate logistic regression analysis to avoid information loss (*Moons et al., 2015*). We also replaced values with missing data with mean imputation rather than excluding indicators, an approach that can lead to a biased effect estimation and decrease the discriminative ability of multivariable models compared with the values (*Moons et al., 2015*).

Our study has the following limitations. First, this was a single-centre retrospective study. Although the medical records and regular laboratory tests were intact, there were still some missing data that could be overfitted by the model, resulting in an overestimation of its performance. Second, the study included haematology indexes such as the MLR, which made the exclusion criteria stricter, and finally included only 398 cases, resulting in a reduction in the number of predictors included in multivariate analysis. For example, we did not include IL-6 and high-sensitivity CRP, which are typical indexes of inflammation. However, we included MLR, a cost-effective inflammation marker, instead. Third, to ensure the accuracy of disease diagnosis, we did not include CVDs that are difficult to diagnose by medical history (such as heart failure that is challenging to distinguish by liquid overload (*Roehm, Gulati & Weiner, 2020*) or diseases that are difficult to define as an event (such as stable angina). Finally, we did not validate the model externally.

## CONCLUSIONS

In the present study, we created a model that can predict the long-term risk of CV events or all-cause mortality in HD patients. Our model comprises only seven variables that are routinely obtained at every dialysis centre and provides a new cost-effective and noninvasive tool to increase early detection and reduce the risk of CV events and mortality. In a future study, we will externally validate the model in prospective cohorts.

## ACKNOWLEDGEMENTS

We thank all medical staff of the Department of Blood Purification who participated in this study.
### Funding
The authors received no funding for this work.

### Competing Interests
The authors declare there are no competing interests.

### Author Contributions

- Aihong Zhang conceived and designed the experiments, performed the experiments, analyzed the data, prepared figures and/or tables, authored or reviewed drafts of the article, and approved the final draft.
- Lemuge Qi performed the experiments, prepared figures and/or tables, and approved the final draft.
- Yanping Zhang performed the experiments, prepared figures and/or tables, and approved the final draft.
- Zhuo Ren performed the experiments, prepared figures and/or tables, and approved the final draft.
- Chen Zhao performed the experiments, prepared figures and/or tables, and approved the final draft.
- Qian Wang performed the experiments, prepared figures and/or tables, and approved the final draft.
- Kaiming Ren performed the experiments, prepared figures and/or tables, and approved the final draft.
- Jiuxu Bai conceived and designed the experiments, authored or reviewed drafts of the article, and approved the final draft.
- Ning Cao conceived and designed the experiments, authored or reviewed drafts of the article, and approved the final draft.

### Human Ethics
The following information was supplied relating to ethical approvals (i.e., approving body and any reference numbers):

The study was approved by the Clinical Research Ethics Committee of the General Hospital of Northern Theater Command.

### Data Availability
The raw measurements are available in the Supplemental File.

### Supplemental Information
Supplemental information for this article can be found online at http://dx.doi.org/10.7717/peerj.14316#supplemental-information.

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
