# Peer review of "Development of a prediction model to estimate the 5-year risk of cardiovascular events and all-cause mortality in haemodialysis patients: a retrospective study"

_PeerJ, doi:10.7717/peerj.14316_

## Round 0.1 · original submission · Major Revisions

More details on the statistical approach, power and validity are needed.

Reviewer 1 ·

Basic reporting

Authors could discuss this aspect in relation to their data:

Since haemodialysis is a lifesaving therapy, adequate control measures are necessary to evaluate its adequacy and to constantly adjust the dose to reduce hospitalisation and prolong patient survival. Malnutrition is common in haemodialysis patients and closely related to morbidity and mortality. Patients undergoing haemodialysis have a high prevalence of1) protein-energy malnutrition and 2)inflammation, along with 3)abnormal iron status. The haemodialysis dose delivered is an important predictor of patient outcome.In keeping with the results of a recent study, authors underline that the use of transferrin levels to assess haemodialysis quality combine into a single test the evaluation of the three most important factors of protein-energy wasting....as evident in.... Do Transferrin Levels Predict Haemodialysis Adequacy in Patients with End-Stage Renal Disease? Nutrients. 2019 May 20;11(5):1123. doi: 10.3390/nu11051123. PMID: 31137583; PMCID: PMC6566169.

How was the collinearity among predictors studied?
Authors should present accurate statistics and obviously data on this point

Experimental design

Good experimental design

Validity of the findings

Findings to be better presented as suggested (collinearity).
Some aspects should be deepened in the Discussion section as suggested.

Additional comments

Authors should be congratulated for approaching an interesting topic.

Reviewer 2 ·

Basic reporting

The research background and motivation is clearly stated in the article. The article structure conforms to PeerJ standards. In general, the English language should be improved throughout the article to ensure that the manuscript is unambiguous, and readers can clearly understand your text. Specific comments below.

1. Line 48. It’s not clear what the authors refer to the “healthy” population. Do you mean “in the general population”?

2. Line 52-56 from “Furthermore, … (ESRD) patients”. I understand what the authors mean there. But the language is vague and should be more specific and clear.

3. Line 117. Using X2 test. Suggest to use Chi-squared test.

4. Line 221-223. It’s not clear the authors mean for “whose dialysis age was more than 6 months because …”. Please clarify and elaborate.

Experimental design

1. The same question has been at least addressed in the Japanese population before, but the title and abstract of this article suggested that this is the first time to develop this kind of prediction model to estimate the risk of CV events and all-cause mortality in HD patients. It would be better to be more specific, for example, referring to Chinese population or from retrospective study in China.

2. This study only included 398 maintenance HD patients, which is a major concern for the validity of the prediction model. The authors included both traditional CV risk factors and more HD-specific risk factors, but are there any risk factors that should be included but not included due to the limited sample size? Please clarify.

3. In the discussion section (line 225-226), the authors mentioned they included patients aged more than 18 years rather than only older dialysis patients in previous studies. What’s the difference between including and not including these younger patients? What’s the implication of this consideration on the performance of the prediction model and the practical use of the model? Please clarify and elaborate.

4. Line 289-290. The authors mentioned a methodological consideration on not including continuous variable with linear relationship. I assume it refers to the variable “Age”. But it’s not clear about the authors’ thoughts here. It’s very common to include age as a continuous variable in the prediction model in practice. Please clarify how the authors address the variable “Age” in the prediction model and what this methodological consideration specifically refers to.

Validity of the findings

1. The Framingham risk score (FRS) is a widely used tool to predict the 10-year cardiovascular disease risk. Did the authors compare the developed risk score with the FRS? It’s important for readers to understand the similarity and difference between the newly developed risk score with a widely used risk score.

2. The authors used item mean imputation. It’s a commonly used and an appropriate method in this context, but did the authors compare the performance of the prediction model before and after imputation? It will be helpful for readers to understand the impact of missing data and imputation method on the prediction model performance as missing values are fairly common in real-world data?

3. Figure 5. the legend is not shown completely in the figure of calibration curve (Figure 5). And Please at least add some notes to help readers understand the legend. What are “apparent”, “bias-corrected” and “ideal” in the figure?

---

## Round 0.2 · accepted · Accept

The authors satisfactorily addressed all the previous suggestions from the reviewers.

Reviewer 1 ·

Basic reporting

The basic reporting is clear

Experimental design

The experimental design is valid

Validity of the findings

The findings are valid

Additional comments

Authors correctly answered comments

Reviewer 2 ·

Basic reporting

The authors have appropriately addressed my comments.

Experimental design

The authors have appropriately addressed my comments.

Validity of the findings

The authors have appropriately addressed my comments.

Additional comments

The authors have appropriately addressed my comments.